# CAGE: Continuity-Aware edGE Network Unlocks Robust Floorplan Reconstruction

**Yiyi Liu**[1]  **Chunyang Liu**[1]  **Bohan Wang**[1]  **Weiqin Jiao**[2]
**Bojian Wu**[3]  **Lubin Fan**[3]  **Yuwei Chen**[4]  **Fashuai Li**[5,*]  **Biao Xiong**[1,*]
[1]Wuhan University of Technology  [2]University of Twente  [3]Independent Researcher
[4]Hangzhou Institute for Advanced Study, University of Chinese Academy of Sciences
[5]The Advanced Laser Technology Laboratory of Anhui Province
*Corresponding Authors: `lifashuai@gmail.com`, `b.xiong@whut.edu.cn`

## Abstract

We present **CAGE** (*Continuity-Aware edGE*) network, a robust framework for reconstructing vector floorplans directly from point-cloud density maps. Traditional corner-based polygon representations are highly sensitive to noise and incomplete observations, often resulting in fragmented or implausible layouts. Recent line grouping methods leverage structural cues to improve robustness but still struggle to recover fine geometric details. To address these limitations, we propose a *native* edge-centric formulation, modeling each wall segment as a directed, geometrically continuous edge. This representation enables inference of coherent floorplan structures, ensuring watertight, topologically valid room boundaries while improving robustness and reducing artifacts. Towards this design, we develop a dual-query transformer decoder that integrates perturbed and latent queries within a denoising framework, which not only stabilizes optimization but also accelerates convergence. Extensive experiments on Structured3D and SceneCAD show that **CAGE** achieves state-of-the-art performance, with F1 scores of 99.1% (rooms), 91.7% (corners), and 89.3% (angles). The method also demonstrates strong cross-dataset generalization, underscoring the efficacy of our architectural innovations. Code and pretrained models are available on our project page: `https://github.com/ee-Liu/CAGE.git`.

## 1 Introduction

Reconstructing indoor scenes into compact, editable vector floorplans is a longstanding goal in computer vision and robotics [1; 2]. A vector floorplan is a structured 2D representation of interior geometry, composed of lines or polygons that delineate walls, room boundaries, and architectural elements. Unlike raster maps, vector floorplans are resolution-independent, support precise geometric reasoning, and integrate seamlessly with CAD tools and building information models (BIM). Their joint encoding of spatial topology and geometry makes them ideal for downstream applications such as building lifecycle management, AR/VR simulation, and autonomous navigation [3; 4; 5; 6].

Modern floorplan reconstruction uses diverse inputs including RGB images [7], panoramic views [8], CAD drawings [9], and 3D point clouds [10]. Among these, projecting point clouds to 2D density maps provides an optimal balance of geometric accuracy and computational efficiency [11; 12], making it popular in learning-based approaches. However, real-world scans often suffer from severe occlusions, clutter, and incompleteness that obscure critical structural elements. Additionally, the discretization process in density map creation blurs corner features and wall segments. These challenges significantly complicate accurate vector floorplan recovery. The resulting incomplete or noisy representations demand robust algorithms capable of handling such imperfect data.

Notably, a critical factor affecting reconstruction quality lies in the choice of floorplan representation. Recent work has explored a range of methods. For example, HEAT [13] detects discrete corners and

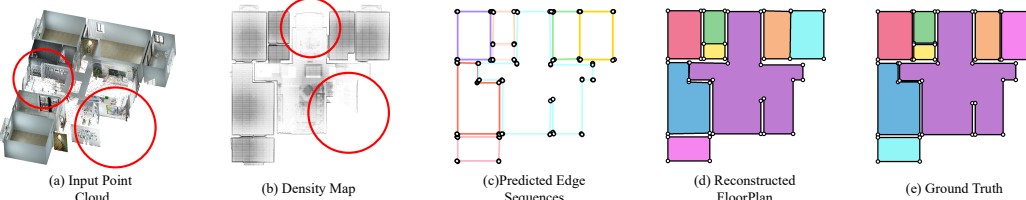

| (a) Input Point Cloud | (b) Density Map | (c) Predicted Edge Sequences | (d) Reconstructed FloorPlan | (e) Ground Truth |

Figure 1: Floorplan reconstruction using our **CAGE** network. Given an input point cloud (a), we project it into a 2D density map (b), predict two-level edge sequences (c), and reconstruct a vector floorplan by intersecting the predicted edges (d). Note that, our edge-based formulation enables the recovery of regular, topologically valid polygons even in severely occluded regions (highlighted with red circles). See Figure 6 for comparisons with corner-based and other SOTA methods.

connects them into edges and polygons, yet it often suffers from gaps caused by missed detections. RoomFormer [12] models floorplans as sequences of corners using a two-level query transformer, yet a single missing corner can distort the entire layout. As a result, corner-based methods are highly sensitive to noise. SLIBO-Net [14] introduces a slicing-box representation but relies heavily on the Manhattan-world assumption. FRI-Net [15] reconstructs rooms by compositional line grouping, its implicit neural representations often lead to over-smoothing of fine structures. It is clear that current forms still struggle to capture global structure while keeping local precision.

To address these challenges, we propose **CAGE** (*Continuity-Aware edGE*) network, a robust framework for reconstructing vector floorplans from density maps. Our **CAGE** adopts an edge-centric formulation, modeling each wall as a directed, geometrically continuous edge. This design improves robustness to incomplete or noisy data by removing reliance on precise corner localization, while enabling global reasoning. To complement this representation, we introduce a dual-query transformer decoder that integrates perturbed and latent edge queries within a denoising framework, enhancing training stability and accelerating convergence. As decoding progresses, edge predictions are iteratively refined, yielding clean and watertight polygons. Evaluations on Structured3D [16] and SceneCAD [17] demonstrate that **CAGE** achieves state-of-the-art performance and generalizes well across datasets. Our main contributions are as follows:

- **Continuity-Aware Edge Representation**: We introduce an edge-based polygon formulation that enhances robustness to incomplete and noisy data, while preserving directional and structural consistency.
- **Dual-Query Transformer Decoder**: Our approach leverages perturbed and latent queries within a denoising framework to stabilize training, refine predictions, and accelerate convergence.
- **Strong Performance and Generalization**: We achieve state-of-the-art performance on multiple benchmarks and generalize effectively across datasets, validating the effectiveness of our continuity-aware edge-based formulation.

## 2 Related Work

Approaches to floorplan reconstruction can be broadly categorized into three groups: traditional methods based on classical geometric or machine learning techniques, stage-wise learning-based pipelines that combine hand-crafted and learned components, and fully end-to-end learning methods.

### 2.1 Traditional Methods

Early work in floor- and room-layout recovery relied on low-level vision cues or geometric heuristics. Some methods extract planes from point clouds and optimize line placements to assemble floorplans [18; 19], while others infer structure from panoramic images [20; 8] or CAD drawings [9; 21; 22]. Techniques such as RANSAC-based plane fitting and piecewise merging have been used to generate compact surface meshes that are later flattened into floorplans [23]. Graph-based formulations often cast the reconstruction as shortest-path or global optimization problems, producing watertight layouts but requiring hand-designed potentials [24; 25]. More recent pipelines focus on handling cluttered LiDAR scans via global optimization over curved surfaces [26] or decomposing scenes into walls and objects for lightweight BIM generation [27]. From a single panorama, Pano2CAD estimates Manhattan room geometry and object pose [28], while successors like PanoFormer [29]

support curved wall recovery via tessellated spherical surfaces. Recent work also explores LoD4 building modeling by fusing interior and exterior cues from SfM images [30]. Traditional methods require small amounts of training data while heavily relying on the design of handcrafted features.

## 2.2 Stage-wise Learning-based Methods

Several methods adopt a hybrid pipeline where learning components are combined with post-processing optimization. FloorNet [31] detects corners in a top-view raster and assembles them via integer programming, while HEAT [13] replaces the solver with a transformer-based corner-pair classifier. Floor-SP [11] and MonteFloor [32] segment rooms using Mask-RCNN [33], refine initial contours through shortest path or Monte Carlo Tree Search [34; 35], and encourage geometric consistency by sharing corners and walls. PolyGraph [36] generates wall points using a cross-guided neural network, forms initial triangles, and applies post-processing to refine polygonal layouts. FloorUSG [19] integrates 2D plane instances with 3D geometry to lift RGB features into structured floorplans. ArrangementNet [10] detects walls and partitions space using wall lines, then leverages an extended GCN to model collinearity and coplanarity among surface patches, improving segmentation in complex scenes. SLIBO-Net [14] introduces a slicing box representation with geometric regularization and post-processing for capturing fine local details. While these stage-wise systems achieve strong performance, they are often sensitive to missing primitives and rely on extensive post-optimization.

## 2.3 End-to-End Learning-based Methods

Recent advances favor end-to-end pipelines that directly predict vector floorplan using transformer-based decoders. RoomFormer [12] introduces a transformer architecture with hierarchical queries to jointly predict multiple corner sequences. Although effective, it suffers from unordered predictions due to random query initialization and incomplete polygons caused by missed corners. PolyRoom [37] enhances this framework by segmenting rooms with Mask-RCNN [33] and initializing RoomFormer-style queries from detected contours. PolyDiffuse [38] formulates floorplan generation as a conditional task and introduces a guided set diffusion model for room representation optimization. These corner-based methods treat floorplan reconstruction as discrete corner regression, often neglecting global shape coherence. FRI-Net [15] addresses this by learning room-wise latent codes and decoding them into lines, which are grouped into polygons using a BSP-Net-inspired [39] grouping strategy. Recent methods have progressed from corner-based to line-grouping approaches, highlighting the importance of structural information. However, they still fail to balance robustness with geometric precision. We address this with edge-centric representation that fundamentally reformulates the problem, enabling accurate, topologically valid reconstructions in an end-to-end trainable framework.

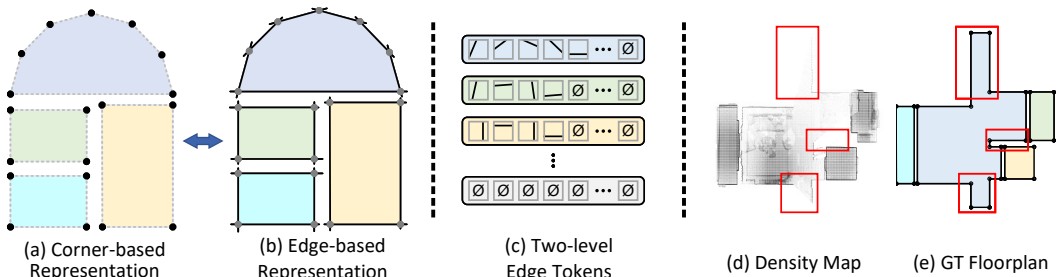

Figure 2: Edge-based Floorplan Representation. (a) Corner-based: polygons defined by sequential vertices; (b) Edge-based: walls represented as directed edges with geometric continuity; (c) Tokenization: polygons represented as sequences of edge tokens at two levels; (d) Density map generated from point cloud; (e) Ground-truth floorplan. While edge and corner representations are mathematically dual, the edge-based formulation offers greater robustness to noise and occlusion (highlighted in red).

## 3 Edge-based Floorplan Representation

We propose an explicit edge-based representation for modeling floorplan, addressing the key limitations of prior methods, see Figure 2(a)(b). We represent each room polygon as an ordered sequence of directed edges, where each edge is defined by a pair of endpoints in normalized 2D space, as illustrated in Figure 2(b). Let a floorplan contain at most $M$ rooms, and each room contain at most $N$

edges. We represent the $m$-th room as:

$$R_m = \{e_m^n\}_{n=1}^{N_m}, \quad \text{where } e_m^n = (\mathbf{p}_{m1}^n, \mathbf{p}_{m2}^n) \tag{1}$$

Each edge $e_m^n \in \mathbb{R}^4$ connects two points $\mathbf{p}_{m1}^n, \mathbf{p}_{m2}^n \in [0,1]^2$. The edge order encodes the polygon's directional traversal. To support batched training and variable-length polygons, we augment each edge with a binary validity label:

$$t_m^n = (\mathbf{p}_{m1}^n, \mathbf{p}_{m2}^n, c_m^n), \quad c_m^n \in \{0, 1\} \tag{2}$$

Here, $c_m^n = 0$ denotes an invalid (padded) edge, and each $t_m^n$ serves as an edge token, as illustrated in Figure 2(c). If all edges in a polygon are invalid, the entire room is treated as padding. While edge and corner representations are dual, edges are empirically more stable in sparse or noisy conditions, since they encode directionality explicitly, promoting angular regularity and structural coherence. Unlike corner-based methods that require precise vertex localization, edge-based design allows endpoints to lie along wall segments. The flexibility simplifies spatial queries and enables the model to infer global structure even from partial observation.

We aggregate edge tokens into a fixed-length set:

$$T_m = \{t_m^n\}_{n=1}^N, \quad \text{where } \max_m N_m \le N \tag{3}$$

and define the full floorplan as:

$$\mathcal{F} = \{T_m\}_{m=1}^M \tag{4}$$

This edge-based representation models room layouts as fixed-length sequences of spatially grounded edge tokens, which align well with transformer architectures, facilitate spatial attention over relevant image regions, and support stable training through bipartite matching.

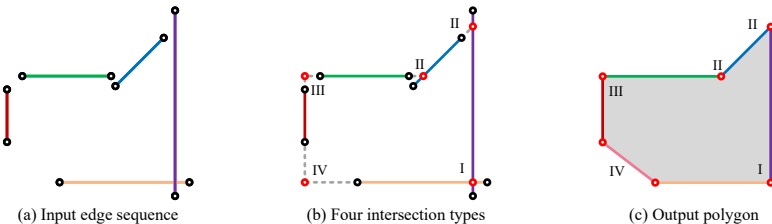

(a) Input edge sequence      (b) Four intersection types      (c) Output polygon

Figure 3: Illustration of edge-to-polygon conversion. (a) Input edge sequence; (b) Four types of edge intersections based on geometric proximity; (c) Final reconstructed polygon.

To convert edge sequences into closed polygons, we resolve edge intersections into vertices. As illustrated in Figure 3, we identify four types of pairwise edge intersections. Types I–III involve intersection points that lie on or near the edges and are retained as valid polygon vertices. In Type IV, where the intersection is far from any endpoints, we instead connect the nearest edge endpoints to preserve continuity. As all edges are ordered and directed, this conversion remains deterministic and robust even under noisy predictions.

## 4 Method

### 4.1 Overview

Figure 4 illustrates the overall architecture of our proposed **CAGE** framework for floorplan reconstruction from 3D point clouds. The input point cloud is first projected onto the XY-plane to generate a 2D density map, which is processed by a convolutional image backbone to extract multi-scale visual features. The output feature maps are flattened and augmented with positional encodings to form a unified token sequence, which is passed to a transformer encoder. We employ multi-scale deformable attention [40] in the encoder to efficiently aggregate both local and global spatial context across all feature levels.

The encoded features are processed by our novel transformer decoder composed of stacked attention layers. The decoder operates on a set of learnable edge queries, which are refined across layers through self-/cross-attention mechanisms. Each query predicts a directed edge denoted by two endpoints and is assigned a binary label indicating whether it constitutes a valid segment, as described in Sec. 3. To improve the robustness and training stability, we incorporate a dual-query design that includes latent queries for final predictions and perturbed queries for denoising supervision. The following subsections will illustrate our novel dual-query decoding mechanism and loss formulation.

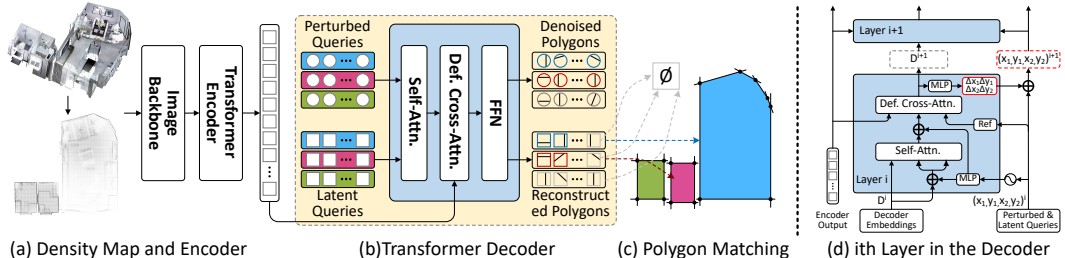

(a) Density Map and Encoder  (b)Transformer Decoder  (c) Polygon Matching  (d) ith Layer in the Decoder

Figure 4: Architecture of the **CAGE** network. (a) The input point cloud is projected into a density map and processed by an image backbone and transformer encoder. (b) The transformer decoder receives two types of queries (perturbed and latent) and predicts edges defined by two endpoints, which may not correspond to polygon vertices. (c) A feed-forward network assigns class labels, with polygon matching for supervision. (d) Detailed architecture of decoder, showing the progressive refinement of edge queries with the incorporation of the novel designs of perturbation and denoising.

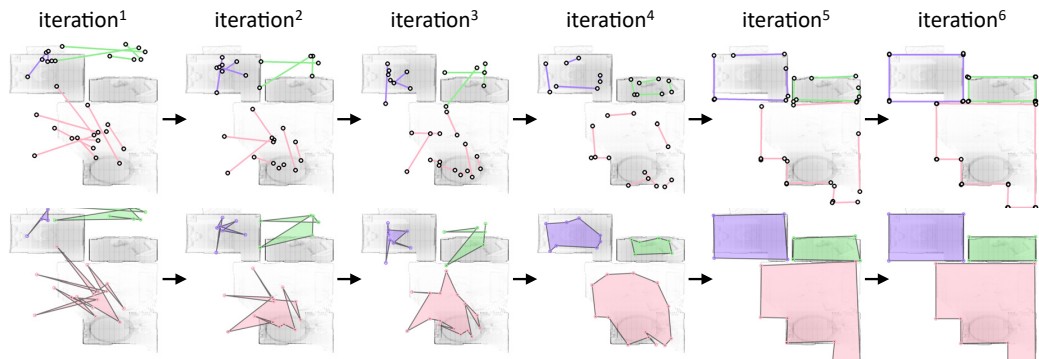

Figure 5: Polygon evolution during decoding. Top: edge query predictions refined over six decoder layers. Bottom: corresponding polygon reconstruction becomes more accurate and complete.

## 4.2 Iterative Polygon Refinement by Dual Query

To enable precise and stable polygon prediction, we design a dual-query transformer decoder that refines edge representations over multiple decoding layers. As shown in Figure 4(d), the decoder jointly operates on two sets of edge queries: perturbed queries used for denoising-based supervision, and latent queries responsible for final polygon prediction. The denoising strategy encourages the model to recover clean edge structures from deliberately corrupted inputs, improving its robustness to noise and accelerating convergence. This dual-query design strengthens learning stability, enhances generalization, and maintains full end-to-end differentiability.

As described in Sec. 3, we represent each room as a sequence of edges and flatten all edges into query tensor as $Q \in \mathbb{R}^{M \times N \times 2}$, where $M$ is the maximum number of polygons and $N$ is the maximum number of edges per polygon. Each query predicts an edge defined by two endpoints. The decoder layers refine these queries iteratively, as illustrated in Figure 5, where edge geometry and corresponding polygons become increasingly accurate across decoding steps.

Each layer receives positional queries derived from polygon coordinates via sinusoidal encoding, along with content queries attending to multi-scale encoder features. Self-attention allows intra-polygon edge refinement and inter-polygon context exchange. Cross-attention modules associate each edge query with spatially grounded encoder features, guided by its predicted endpoint coordinates. This supports effective reasoning over both local structure and global context. A binary classifier identifies edge validity, allowing the model to handle variable polygon structures.

Following Deformable DETR [40], we refine edge coordinates through iterative offset prediction. Each decoder layer outputs coordinate offsets that adjust the endpoints of every edge query, improving geometric precision across layers. Latent queries are randomly initialized and optimized for prediction, while perturbed queries are derived by adding controlled noise to ground-truth edge coordinates. These two query types are processed jointly in the decoder.

To support denoising supervision, we adopt a training objective inspired by DN-DETR [41]. The decoder receives perturbed queries generated by adding controlled noise to ground-truth edges and optionally flipping their class labels. Let the perturbed query set be $\mathbf{q} = \{q_0, \ldots, q_{MN-1}\}$, and the latent query set be $\mathbf{Q} = \{Q_0, \ldots, Q_{MN-1}\}$. The decoder operates on both sets under an attention mask $\mathbf{A}$ that prevents leakage between perturbed instances:

$$\mathbf{o} = D(\mathbf{q}, \mathbf{Q}, \mathbf{F} \mid \mathbf{A}) \tag{5}$$

where $\mathbf{F}$ denotes encoder features.

Each perturbed edge query is generated by applying controlled noise to a ground-truth edge during training. Specifically, for each edge, its two endpoints are randomly displaced to simulate input uncertainty. The displacements are constrained relative to the spatial extent of the density map: the horizontal shift $\Delta x$ and vertical shift $\Delta y$ are bounded by $|\Delta x| < \lambda w/2$ and $|\Delta y| < \lambda h/2$, where $w$ and $h$ denote the width and height of the input density image, respectively. If a perturbed endpoint falls outside the image boundary, it is clamped to remain within the valid range. The scalar $\lambda$ is a hyperparameter that determines the magnitude of perturbation. Additionally, to introduce label noise, the binary class label of each edge is randomly flipped with probability $\gamma$. This denoising strategy encourages the decoder to recover clean edge structures from noisy inputs. During inference, only the latent queries are used to generate final polygons.

### 4.3 Loss Functions

To train the **CAGE** network in an end-to-end manner, we formulate a loss that supervises both the structure and geometry of predicted polygons. The decoder outputs a fixed number $M$ of polygons, each consisting of up to $N$ directed edges. Since the number and order of edges vary across ground-truth annotations, we adopt a two-level matching strategy that aligns predictions with targets.

**Polygon Matching.** Let the predicted edge set be $\hat{S} = \{\hat{E}_m = (\hat{e}_m^1, \ldots, \hat{e}_m^N)\}_{m=1}^M$, where each edge $\hat{e}_m^n = (\hat{c}_m^n, \hat{p}_{m1}^n, \hat{p}_{m2}^n)$ includes a binary confidence score and two endpoint coordinates. Ground-truth polygons are denoted as $S = \{E_m\}_{m=1}^{M_{\text{gt}}}$, each padded to length $N$ and further extended with mock polygons for matching. We compute a bipartite matching between predicted and ground-truth polygons using the Hungarian algorithm, minimizing the following assignment cost:

$$\hat{\sigma} = \arg \min_{\sigma} \sum_{m=1}^{M} \mathcal{D}(E_m, \hat{E}_{\sigma(m)}) \tag{6}$$

where the matching cost $\mathcal{D}$ includes both classification error and edge regression:

$$\mathcal{D}(E_m, \hat{E}_{\sigma(m)}) = \mathbb{1}_{\{m \le M_{\text{gt}}\}} \left[ \lambda_{\text{cls}} \sum_{n=1}^{N} \left| c_m^n - \hat{c}_{\sigma(m)}^n \right| + \sum_{n=1}^{N} \sum_{k=1}^{2} \left\| p_{mk}^n - \hat{p}_{\sigma(mk)}^n \right\|_1 \right] \tag{7}$$

To account for the cyclic nature of polygons, we evaluate distances over all valid rotations of the ground-truth sequence and choose the minimal one.

**Loss Components.** After matching, we supervise the predicted polygons using three terms: classification, edge regression, and rasterization. For latent queries, losses are applied only to matched predictions. For *perturbed queries*, losses are directly applied without the matching step, as their correspondence to ground-truth edges is already known.

The classification loss uses binary cross-entropy:

$$\mathcal{L}_{\text{cls}}^m = -\frac{1}{N} \sum_{n=1}^{N} \left[ c_n^m \log(\hat{c}_n^{\hat{\sigma}(m)}) + (1 - c_n^m) \log(1 - \hat{c}_n^{\hat{\sigma}(m)}) \right] \tag{8}$$

The edge regression loss is the $\ell_1$ distance between predicted and ground-truth endpoints:

$$\mathcal{L}_{\text{edge}}^m = \frac{1}{N_m} \mathbb{1}_{\{m \le M_{\text{gt}}\}} \sum_{n=1}^{N_m} \sum_{k=1}^{2} \left\| p_{mk}^n - \hat{p}_{\hat{\sigma}(mk)}^n \right\|_1 \tag{9}$$

The rasterization loss measures the overlap between polygon masks using the Dice loss [42]:

$$\mathcal{L}_{\text{ras}}^m = \mathbb{1}_{\{m \le M_{\text{gt}}\}} \cdot \text{Dice}\left( R(P_m), R(\hat{P}_{\hat{\sigma}(m)}) \right) \tag{10}$$

where $R(\cdot)$ denotes a differentiable polygon rasterizer [43]

For perturbed queries, we apply denoising losses without matching. These include binary cross-entropy for classification and $\ell_1$ regression for edge localization, denoted as $\mathcal{L}_{\text{cls\_DN}}$ and $\mathcal{L}_{\text{edge\_DN}}$.

The total training loss is the weighted sum of all components:

$$\mathcal{L} = \sum_{m=1}^{M} \left( \lambda_{\text{cls}} \mathcal{L}_{\text{cls}}^m + \lambda_{\text{edge}} \mathcal{L}_{\text{edge}}^m + \lambda_{\text{ras}} \mathcal{L}_{\text{ras}}^m + \lambda_{\text{cls\_DN}} \mathcal{L}_{\text{cls\_DN}}^m + \lambda_{\text{edge\_DN}} \mathcal{L}_{\text{edge\_DN}}^m \right) \qquad (11)$$

## 5 Experiments

### 5.1 Experiments Setting

**Dataset and Evaluation Metrics.** We evaluate our method on two large-scale indoor datasets, including Structured3D [16] and SceneCAD [17]. Structured3D is a photo-realistic synthetic dataset comprising 3,500 houses with diverse Manhattan and non-Manhattan layouts. SceneCAD provides 3D room layout annotations for real-world RGB-D scans from ScanNet [44], with each sample containing a single room. Following previous work [13; 12], we split Structured3D into 3,000 training, 250 validation, and 250 test samples. For SceneCAD, we use the provided splits of 828 training and 127 validation samples. Following [32; 13], we process registered multi-view RGB-D panoramas into point clouds, and project them vertically into $256 \times 256$ pixel density images. Each pixel value is normalized to $[0, 1]$ by dividing the number of projected points by the maximum count per image. In line with prior work [13; 12; 36], we evaluate performance using Precision, Recall, and F1 scores at the Room, Corner, and Angle levels.

**Baselines.** We compare our method with eight state-of-the-art approaches: Floor-SP [11], HEAT [13], RoomFormer [12], SLIBO-Net [14], FRI-Net [15], PolyRoom [37], PolyGraph [36], and PolyDiffuse [38]. Floor-SP [11] uses Mask-RCNN for room segmentation followed by classical optimization for vectorization. HEAT [13] detects corners and infers connectivity with a neural network. RoomFormer [12] employs a transformer to jointly predict multiple corner sequences. SLIBO-Net [14] generates slicing boxes and room centroids via transformer decoding, refined through post-processing. FRI-Net [15] encodes room-wise latent features, decodes them into lines, and assembles polygons using a BSP-Net-inspired [39] grouping strategy. PolyRoom [37] combines Mask-RCNN-based segmentation with RoomFormer-style reconstruction initialized from detected contours. PolyGraph [36] predicts wall points and triangulations and then refines results by postprocessing. PolyDiffuse [38] formulates reconstruction as a conditional generation task, using a diffusion model to refine floorplans from initial proposals iteratively.

**Implementation Details.** Our model is implemented in PyTorch and trained on a single NVIDIA RTX 4090 GPU. We use Swin Transformer V2 [45] as the default image backbone, while ResNet-50 [46] and Swin Transformer V1 [47] are used in ablation studies. The transformer architecture follows a standard 6-layer encoder-decoder configuration. We apply denoising training with dual-query supervision and optimize using Adam [48]. Further model configurations, training schedules, and hyperparameters are provided in supplementary material A.

### 5.2 Comparisons to State-of-the-Art Methods

Table 1: Quantitative comparison on the Structured3D [16] test set. The running time is averaged over the whole test set. PD denotes PolyDiffuse [38] post-optimization. *: Params are calculated from the officially released models.

| Method | Venue | Backbone | Params(M)* | t (s) | Room | | | Corner | | | Angle | | |
|---|---|---|---|---|---|---|---|---|---|---|---|---|---|
| | | | | | Prec. | Rec. | F1 | Prec. | Rec. | F1 | Prec. | Rec. | F1 |
| HEAT [13] | CVPR22 | ResNet-50 | 48.9 | 0.11 | 96.9 | 94.0 | 95.4 | 81.7 | 83.2 | 82.5 | 77.6 | 79.0 | 78.3 |
| RoomFormer [12] | CVPR23 | ResNet-50 | 40.6 | 0.01 | 97.9 | 96.7 | 97.3 | 89.2 | 85.3 | 87.2 | 83.0 | 79.6 | 81.3 |
| SLIBO-Net [14] | NeurIPS23 | ResNet-50 | – | 0.17 | 99.1 | 97.8 | 98.4 | 88.9 | 82.1 | 85.4 | 87.8 | 81.2 | 84.4 |
| PolyRoom [37] | ECCV24 | ResNet-50 | 42.2 | 0.02 | 98.9 | 97.7 | 98.3 | 94.6 | 86.1 | 90.2 | 89.3 | 81.4 | 85.2 |
| FRI-Net [15] | ECCV24 | ResNet-50 | 58.6 | 0.09 | 99.5 | **98.7** | **99.1** | 90.8 | 84.9 | 87.8 | 89.6 | 84.3 | 86.9 |
| PolyGraph [36] | TVCG25 | ResNet-50 | – | 0.04 | 95.7 | 97.9 | 96.7 | 92.4 | 82.2 | 88.3 | 89.2 | 79.4 | 85.4 |
| CAGE(Our) | – | SwinV2-L | 211.3 | 0.01 | **99.6** | **98.7** | **99.1** | 95.0 | 88.6 | 91.7 | 92.5 | 86.4 | 89.3 |
| RoomFormer+PD [38] | NeurIPS23 | ResNet-50 | – | – | 98.7 | 98.1 | 98.4 | 92.8 | 89.3 | 91.0 | 90.8 | 87.4 | 89.1 |
| FRI-Net+PD [15] | ECCV24 | ResNet-50 | – | – | 99.6 | 98.6 | **99.1** | 94.2 | 88.2 | 91.1 | **91.9** | 86.7 | **89.2** |
| Ours+PD | – | SwinV2-L | – | – | **99.7** | **98.7** | **99.1** | **95.1** | **89.9** | 91.7 | 91.6 | **87.5** | **89.2** |

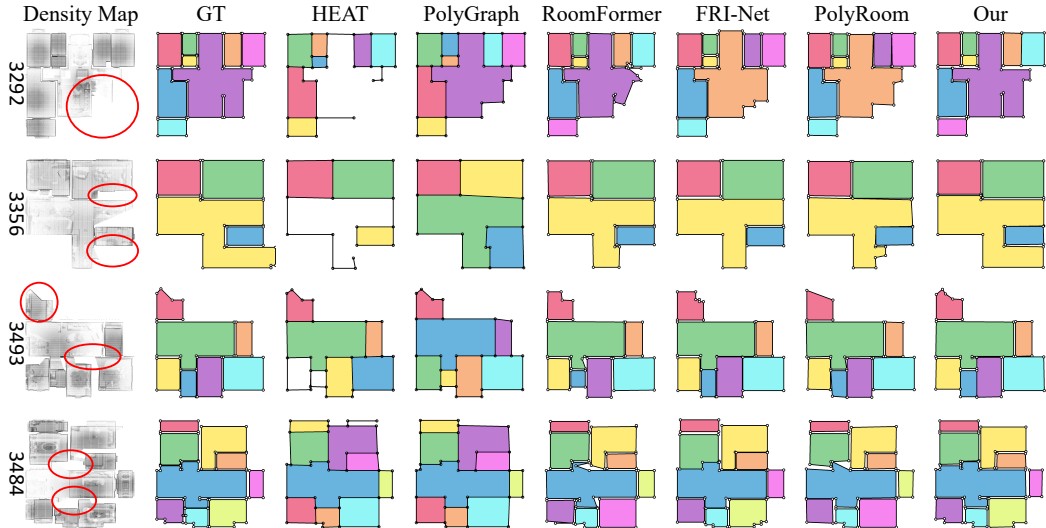

Figure 6: Qualitative comparisons on Structured3D [16]. Scene IDs are shown on the left. Red circles highlight challenging regions.

**Quantitative Evaluation.** Quantitative comparisons against state-of-the-art floorplan reconstruction methods on the Structured3D [16] dataset are presented in Table 1. We compare our method with HEAT [13], SLIBO-Net [14], PolyRoom [37], FRI-Net [15], and PolyGraph [36]. For completeness, we also include post-optimization results using PolyDiffuse [38]. Without any post-processing, our method achieves the best performance across all geometric levels, reaching 99.1% Room F1, 91.7% Corner F1, and 89.3% Angle F1. Although FRI-Net attains comparable performance in Room Recall and Room F1, **CAGE** surpasses it by a notable margin of +3.9 points in Corner F1 and +2.4 points in Angle F1, demonstrating superior accuracy in fine-grained corner localization and angular estimation.

After applying PolyDiffuse for post-optimization, our method further consolidates its leading position across all three evaluation levels, outperforming both RoomFormer and FRI-Net. Interestingly, the post-optimization with PolyDiffuse brings only marginal improvements to our model, indicating that **CAGE** already captures the scene structure effectively without reliance on heavy post-processing. In contrast, PolyDiffuse, as a diffusion-based model, requires substantial additional training time. Furthermore, despite not explicitly optimizing for corner predictions, our method maintains competitive performance in Corner Recall. Regarding efficiency, **CAGE** achieves the fastest inference speed, along with RoomFormer (both 0.01 second per image), benefiting from its lightweight architecture without the need for additional refinement stages. Overall, these results demonstrate that **CAGE** strikes an excellent balance between reconstruction accuracy and computational efficiency.

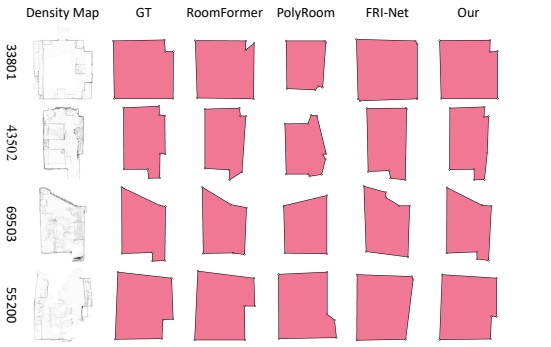

Figure 7: Qualitative evaluations on SceneCAD [17].

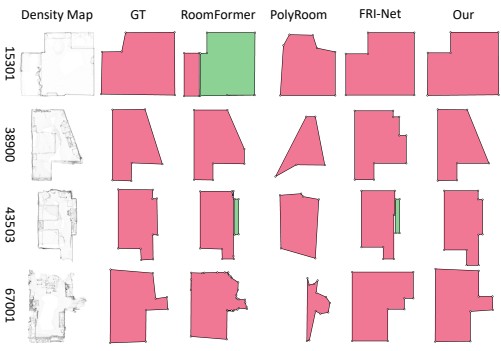

Figure 8: Cross-dataset evaluation (training on Structured3D [16] and testing on SceneCAD [17]).

On the SceneCAD [17] dataset, we further evaluate our method against Floor-SP [11], HEAT [13], RoomFormer [12], FRI-Net [15], and PolyRoom [37]. Given that SceneCAD primarily consists of single-room layouts, we assess reconstruction quality using IoU for room shape. As shown in Table 2, **CAGE** consistently achieves either the best or second-best performance across nearly all evaluation metrics. Specifically, our method attains the highest scores in Room IoU, Corner Recall, Angle Recall, and Angle F1, while securing second-best results in other metrics. In comparison, PolyRoom achieves the highest Corner Precision and Corner F1 but performs slightly lower on remaining metrics. Overall, **CAGE** demonstrates strong and balanced performance across all aspects of room layout reconstruction on the SceneCAD dataset.

**Qualitative Evaluation.** Visual comparisons on Structured3D [16] are shown in Figure 6, with red circles highlighting challenging regions for reconstruction. Most state-of-the-art methods fail to recover severely occluded or blocked areas, while our method successfully infers structures using weak wall cues (e.g., scene 3292). HEAT [13] and PolyGraph [36] reconstruct wall centerlines, whereas other methods produce double-wall representations, with each polygon denoting room interior boundaries. HEAT follows a two-stage process—detecting corners and inferring connectivity—which often leads to missing corners and incorrect edge associations (e.g., scene 3493), resulting in open polygons. PolyRoom [37] generates simple, closed polygons but struggles in occluded regions (e.g., 3292, 3356). RoomFormer [12] predicts polygonal layouts via sequences of corners, ensuring closure but often producing irregular shapes due to weak constraints on edge relationships (e.g., 3292, 3493, 3484). While FRI-Net [15] and PolyRoom demonstrate better regularity and partial robustness to occlusion, they still fail under heavy blockage (e.g., 3292, 3356). In contrast, our method explicitly models global structure through edge-aware representation and denoising-based training, leading to more regular room geometries and robust reconstruction in occluded scenarios. Figure 7 further confirms this trend, with our model consistently producing more plausible results compared to corner-based optimization methods.

**Cross-Dataset Generalization.** In cross-dataset evaluation (training on Structured3D [16] and testing on SceneCAD [17]), *PolyRoom* and **CAGE** achieve the top results across Room IoU and fine-level corner/angle metrics (Table 3). PolyRoom's segmentation-based pipeline aligns well with SceneCAD's single-room layouts, yielding strong room-level performance, but its multi-stage design struggles with the complex multi-room structures in Structured3D (Table 1). In contrast, **CAGE**'s end-to-end trainable, edge-based modeling consistently maintains high accuracy (85.6% Room IoU, 87.5% Corner Recall, 70.6% Angle Recall, 61.6% Angle F1) without post-processing. These results validate that combining edge-based representations with global shape modeling and denoising strategies enables **CAGE** to achieve robust and balanced floorplan reconstruction under domain shifts and varying scene complexities.

Table 2: Quantitative comparison on the SceneCAD validation set [17]. Results for prior methods are reported from [12; 15; 38].

| Method | Room | Corner | | | Angle | | |
|---|---|---|---|---|---|---|---|
| | IOU | Prec. | Rec. | F1 | Prec. | Rec. | F1 |
| Floor-SP [11] | 91.6 | 89.4 | 85.8 | 87.6 | 74.3 | 71.9 | 73.1 |
| HEAT [13] | 84.9 | 87.8 | 79.1 | 83.2 | 73.2 | 67.8 | 70.4 |
| RoomFormer [12] | 91.7 | 92.5 | 85.3 | 88.8 | 78.0 | 73.7 | 75.8 |
| FRI-Net [15] | 92.3 | 92.8 | 85.9 | 89.2 | 78.3 | 73.6 | 75.9 |
| PolyRoom [37] | 92.8 | **96.8** | 86.1 | **91.2** | **81.7** | 74.5 | 78.0 |
| CAGE(Ours) | **93.7** | 93.7 | **87.7** | 90.6 | 81.2 | **77.3** | **79.2** |

Table 3: Cross-data generalization. Models are trained on Structured3D train set but evaluated on SceneCAD val set.

| Method | Room | Corner | | | Angle | | |
|---|---|---|---|---|---|---|---|
| | IOU | Prec. | Rec. | F1 | Prec. | Rec. | F1 |
| HEAT [13] | 52.5 | 50.9 | 51.1 | 51.0 | 42.2 | 42.0 | 41.6 |
| RoomFormer [12] | 74.0 | 56.2 | 65.0 | 60.3 | 44.2 | 48.4 | 46.2 |
| FRI-Net [15] | 80.6 | 66.4 | 79.5 | 72.4 | 56.3 | 67.2 | 61.3 |
| PolyRoom [37] | 85.2 | **77.8** | 79.9 | **78.9** | **59.4** | 61.7 | 60.6 |
| CAGE (Ours) | **85.6** | 68.2 | **87.5** | 76.7 | 54.6 | **70.6** | **61.6** |

## 5.3 Ablation Study

We evaluate the impact of three core components: the edge-based polygon representation, the denoising (DN) training strategy, and the image backbone. As shown in Table 4, both the Edge and DN modules yield consistent improvements individually, and their combination results in further performance gains. By contrast, incorporating SwinTransformer-V2 as the backbone provides only marginal benefits beyond the combined Edge and DN configuration. Further analysis and visual examples are provided in Supplementary Material B.5.

**Polygon Representation via Edge.** We first examine the effect of the proposed Edge module, as shown in the third row of Table 4. In this variant, the original corner-based representation in RoomFormer is replaced with an edge-based formulation, while keeping all other settings unchanged.

Table 4: Ablation study on Structured3D. Edge and DN modules significantly improve performance. SwinT backbones provide limited or reduced gains compared to ResNet50. RoomFormer [12] results are reproduced using our training setup for fair comparison.

| Method | Edge | DN | Backbone | Room | | | Corner | | | Angle | | |
|--------|------|----|----------|------|------|----|--------|------|----|-------|------|----|
| | | | | Prec. | Rec. | F1 | Prec. | Rec. | F1 | Prec. | Rec. | F1 |
| RoomFormer | – | – | ResNet50[46] | 96.9 | 95.9 | 96.4 | 87.3 | 83.8 | 85.5 | 81.2 | 77.9 | 79.5 |
| Ours | ✓ | – | ResNet50[46] | 98.7 | 97.6 | 98.2 | 91.2 | 87.0 | 89.0 | 85.1 | 81.3 | 83.2 |
| Ours | – | ✓ | ResNet50[46] | 98.0 | 96.2 | 97.1 | 90.4 | 84.4 | 87.3 | 85.0 | 79.5 | 82.2 |
| Ours | – | – | SwinT_V2[45] | 97.0 | 95.7 | 96.3 | 87.3 | 83.4 | 85.3 | 80.5 | 76.9 | 78.6 |
| Ours | ✓ | ✓ | ResNet50[46] | **99.6** | **98.7** | **99.2** | 93.9 | 88.8 | 91.3 | 89.7 | 84.9 | 87.2 |
| Ours | ✓ | ✓ | SwinT_V1[47] | 99.4 | 98.5 | 98.9 | 94.5 | **89.1** | **91.7** | 91.1 | 85.9 | 88.4 |
| Ours | ✓ | ✓ | SwinT_V2[45] | **99.6** | **98.7** | 99.1 | **95.0** | 88.6 | **91.7** | **92.5** | **86.4** | **89.3** |

The Edge module consistently improves performance across all evaluation metrics, increasing Room F1 from 96.4% to 98.2%, Corner F1 from 85.5% to 89.0%, and Angle F1 from 79.5% to 83.2%. These results demonstrate that representing polygons by edges enables the model to more effectively capture structural geometry and enhances its ability to localize wall boundaries.

**Training Strategy via Denoising.** To further improve learning stability and geometric reasoning, we incorporate a denoising strategy in which ground-truth data with noise are introduced alongside random queries during training. This approach supports convergence in the early stages of prediction and stablize the Hungarian Matching process. As illustrated in the third row of Table 4, the addition of the DN module alone leads to moderate gains. While the DN module is less impactful than the Edge module in isolation, its value becomes more evident in combination.

When both Edge and DN modules are integrated (fifth row of Table 4), we observe substantial performance gains across all metrics, with Room F1 rising to 99.2%, Corner F1 to 91.3%, and Angle F1 to 87.2%. Notably, the most significant gains are observed in the Corner and Angle metrics, indicating improved precision in local geometric structures. This synergy suggests that the Edge and DN modules complement each other by jointly enhancing both global room layout understanding and fine-grained structural details.

**Image Backbone.** We evaluate the impact of different image backbones—ResNet-50 [46], SwinTransformer-V1 [47], and SwinTransformer-V2 [45]—while keeping the proposed Edge and DN modules fixed. All configurations utilize four image scales and a token dimension of 256. As shown in the last three rows of Table 4, integrating Swin Transformer backbones yields mixed results. While SwinTransformer-V2 achieves the highest Angle F1 score (89.3%) and slightly improves Corner F1 (91.7%), these gains are relatively modest compared to the improvements brought by the Edge and DN modules. Notably, ResNet-50 combined with both modules attains the highest Room F1 score (99.2%), underscoring its effectiveness in capturing global structural context.

These results suggest that the primary performance improvements stem from the proposed architectural enhancements rather than the choice of backbone. SwinTransformer variants offer benefits in modeling local geometric relationships through shifted windowing and cross-boundary attention, particularly enhancing the Corner and Angle metrics. However, the convolutional features of ResNet-50 remain highly effective for global layout prediction. Overall, SwinTransformer-V2 provides a balanced trade-off between local and global representation capabilities. Therefore, unless otherwise stated, we adopt the configuration with Edge, DN, and SwinTransformer-V2 as the default model for subsequent experiments.

# 6 Conclusion

We present **CAGE**, an end-to-end floorplan reconstruction framework that models rooms as sequences of edges using a dual-query transformer architecture. By predicting edges instead of corners, our method captures global geometric structure and directional priors, enabling robust inference of room layouts even from partially observed wall segments through edge-based attention. This significantly improves resilience to occlusion and layout noise. A denoising-based training strategy with perturbed and latent queries facilitates stable and efficient iterative refinement. Extensive experiments on Structured3D and SceneCAD demonstrate state-of-the-art performance. Our method still puzzles over reconstructing scenes with dense occlusions or fine structural elements. Future work includes integrating semantic priors, extending to full architectural elements, and scaling to outdoor scenarios.

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

# Supplementary Materials

## A  Implementation Details

**Model Settings.** Our primary image backbone is Swin Transformer V2 [45], while ResNet-50 [46] and Swin Transformer V1 [47] are used for comparison in the ablation study. Following Room-Former [12], for ResNet-50, we extract multi-scale feature maps from the last three stages without using a feature pyramid network (FPN). For Swin Transformers, we adopt the official large-variant configurations: SwinV1 uses Swin-L pretrained weights, and SwinV2 uses SwinV2-L pretrained weights, both with a base dimension of $C = 192$ and stage depths of [2, 2, 18, 2]. The window sizes differ, with SwinV1 using a window size of 12 and SwinV2 using 16. For both ResNet-50 and Swin Transformer backbones, the fourth-scale feature map is obtained via a $3 \times 3$ convolution with stride 2 applied to the final-stage output. All feature maps are projected to 256 channels before being fed into the transformer. Pretrained weights from [46; 47; 45] are used for all backbone layers and are fine-tuned during training. Our transformer architecture comprises 6 encoder layers and 6 decoder layers, each with a hidden dimension of 256. We set the number of room feature codes $m$ to 20 and the number of edge queries $n$ to 40.

**Training.** We train our model using the Adam optimizer [48] with a weight decay of $1e^{-4}$. Depending on the dataset size, we train the model on Structured3D for 650 epochs with an initial learning rate of $2e^{-4}$, and on SceneCAD for 400 epochs with an initial learning rate of $5e^{-5}$. In both cases, the learning rate is decayed by a factor of 0.1 during the final 20% of epochs. The loss weights are set as $\lambda_{cls} = 0.6$, $\lambda_{edge} = 6$, $\lambda_{ras} = 1$, $\lambda_{DN\_cls} = 0.6$, and $\lambda_{DN\_edge} = 6$. For perturbed queries, the noise scale is set to 0.2 for class and 0.4 for edge. We implement our model in PyTorch and conduct all experiments on an NVIDIA GeForce RTX 4090 GPU with 24 GB of memory.

# B  Additional Results and Visualizations

## B.1  Evolution of Edge Queries

To further examine the decoding process, we visulize a layer-wise edge query refinement as illustrated in Figure 9. As decoding progresses, the predicted edge structures become increasingly coherent and better aligned with the underlying point density map, even in regions with limited or missing wall evidence. Correspondingly, the reconstructed polygons evolve from coarse outlines to precise, watertight layouts. This illustrates the effectiveness of our progressive decoding strategy in improving both geometric accuracy and topological consistency.

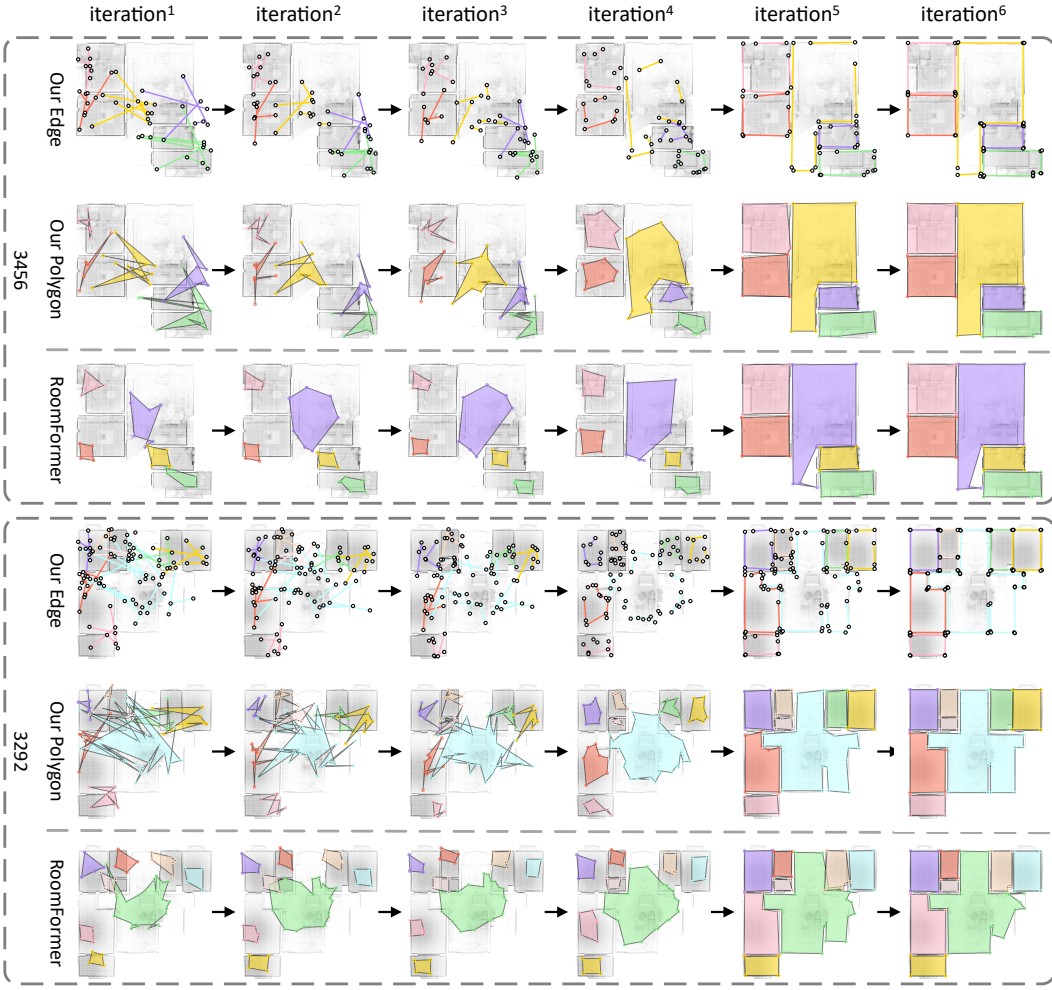

Figure 9: Additional results for edge query evolution and polygon formation. Each example shows a density map from Structure3D [16] (scene ID on the left), with edge query predictions over six decoding layers (top row), the corresponding polygon reconstructions (second row), and the polygon evolution of Roomformer [12] (third row).

## B.2 More Reconstruction result on Structured3D

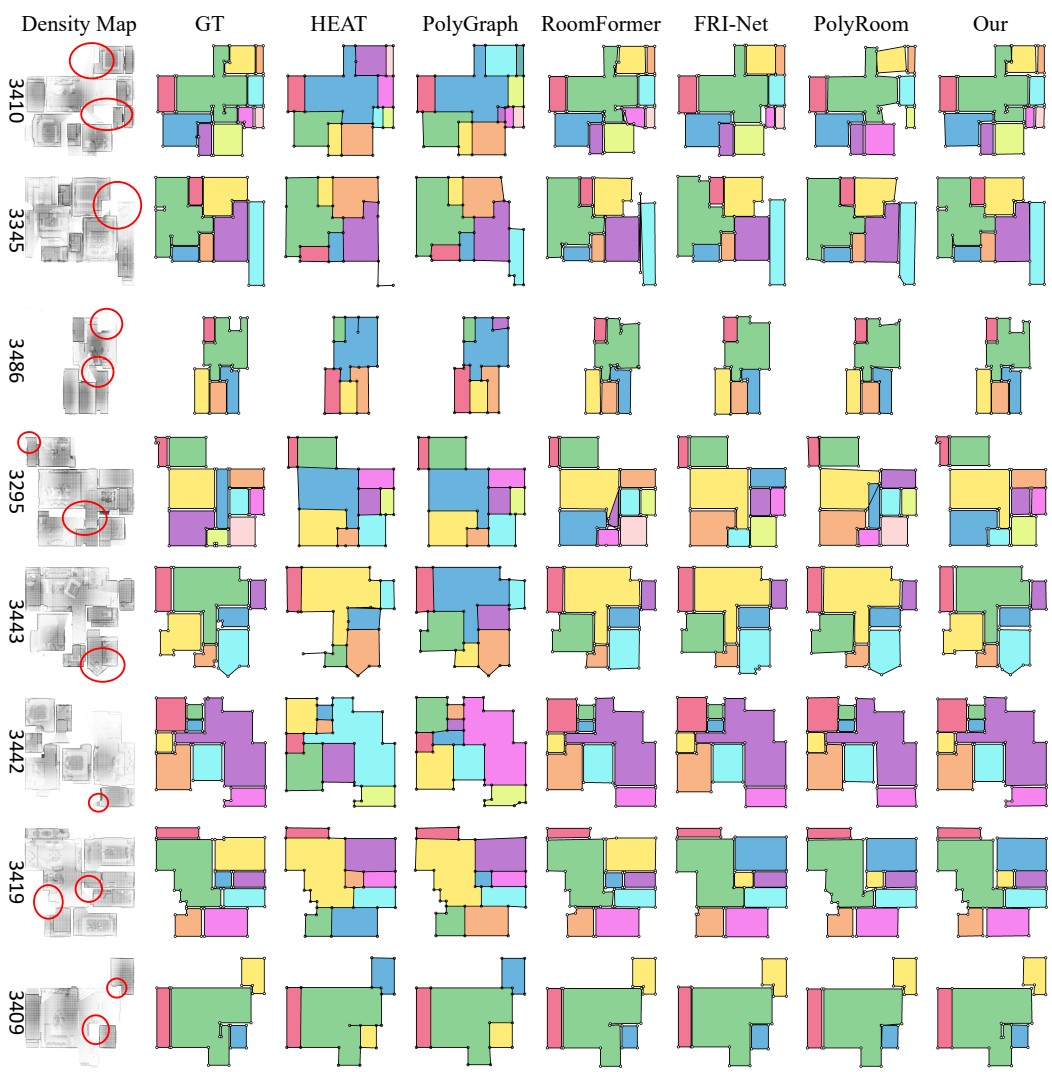

Figure 10: Additional results for Qualitative evaluations on Structured3D [17].

## B.3 Failure cases on Structure3D

Our proposed **CAGE** model demonstrates strong performance on both the Structure3D and SceneCAD datasets. However, several typical failure cases remain, as illustrated in Fig. 11. In some cases, ambiguity in corner completion arises: certain ground truth layouts require polygonal closure at missing corners, while others do not, leading to inconsistencies in prediction (Fig. 11(a)). Additionally, small architectural elements such as interior rooms or columns may be absent from the input density map, resulting in missed detections (Fig. 11(b)). Severe occlusions caused by missing scan stations can further result in incomplete reconstructions (Fig. 11(c)). Lastly, we observe variability in prediction quality across different checkpoints within the same training run—some checkpoints yield strong results, while others perform poorly—likely due to sensitivity to parameter initialization or optimization instability (Fig. 11(d)). Addressing these challenges may involve incorporating more robust data augmentation techniques, exploring reinforcement learning-based refinement, or leveraging checkpoint ensembles to enhance model stability and consistency.

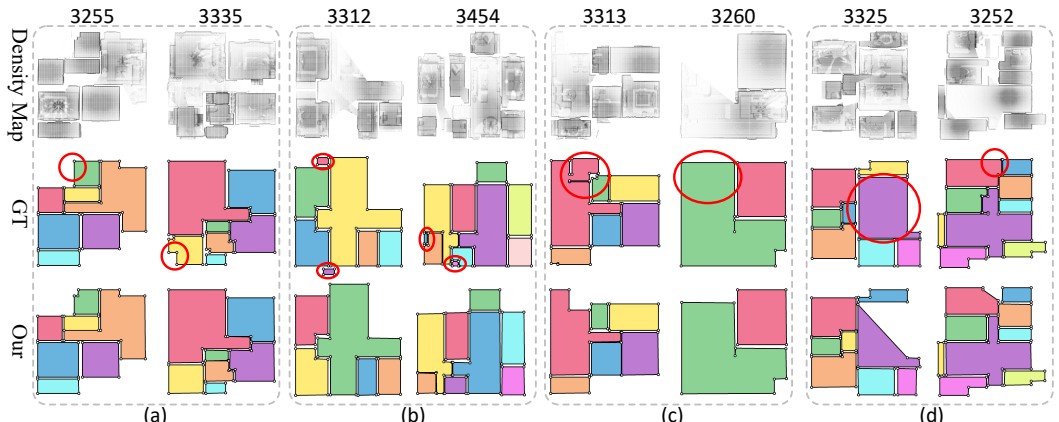

Figure 11: Failure cases on Structure3D [16], including polygon ambiguity(a), missing structures(b), incomplete scans(c), and checkpoint instability(d).

## B.4 More Reconstruction result on SceneCAD

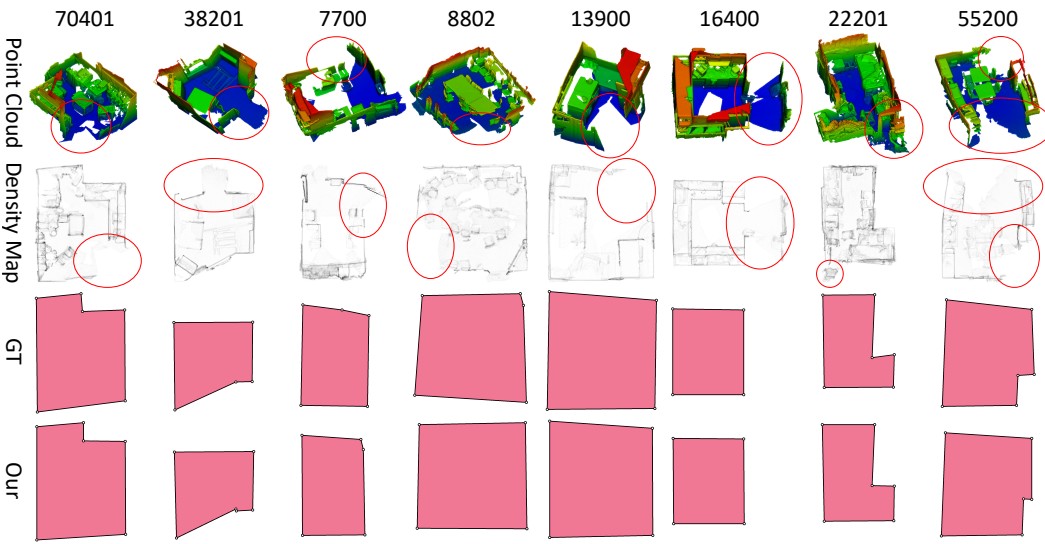

Figure 12: Additional results for Qualitative evaluations on SceneCAD [17].

Table 5: Sensitivity of reconstruction accuracy to threshold values. †: the threshold we used in our experiments.

| Eps | Room | | | Corner | | | Angle | | |
|---|---|---|---|---|---|---|---|---|---|
| | Prec. | Rec. | F1 | Prec. | Rec. | F1 | Prec. | Rec. | F1 |
| 0.2 | 99.59 | 98.69 | 99.14 | 95.04 | 88.57 | 91.70 | 92.57 | 86.40 | 89.39 |
| 0.1† | 99.59 | 98.69 | 99.14 | 94.99 | 88.55 | 91.66 | 92.51 | 86.37 | 89.34 |
| 0.05 | 99.59 | 98.69 | 99.14 | 94.96 | 88.55 | 91.65 | 92.46 | 86.34 | 89.30 |
| 0.01 | 99.59 | 98.69 | 99.14 | 94.95 | 88.58 | 91.66 | 92.40 | 86.34 | 89.27 |
| 0.0001 | 99.59 | 98.69 | 99.14 | 94.94 | 88.60 | 91.67 | 92.38 | 86.34 | 89.26 |

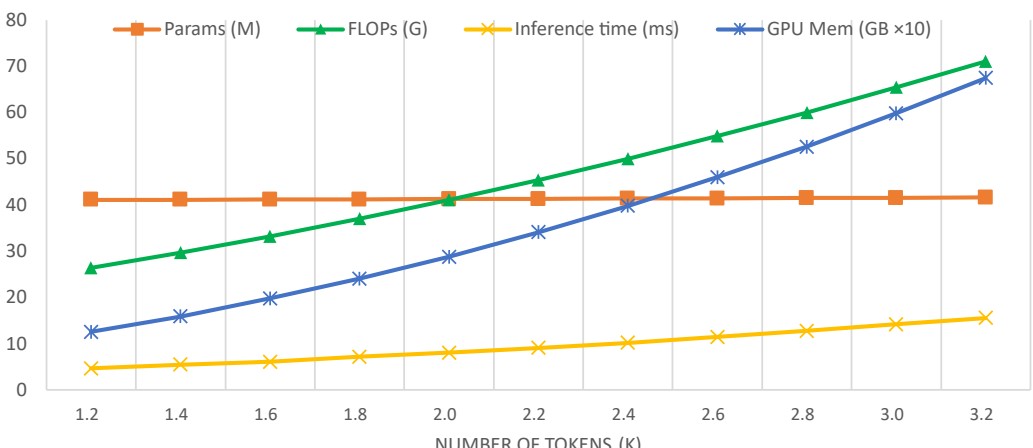

Figure 13: Impact of the number of tokens on model complexity and efficiency. As the token count increases, the number of parameters, FLOPs, inference time, and GPU memory usage grow nearly quadratically, reflecting the scalability characteristics of transformer-based architectures.

## B.5 Extended Evaluation

**Sensitivity to threshold in edge-to-polygon conversion.** The edge-to-polygon conversion's sensitivity to the threshold (epsilon) is minimal as shown in Table 5, varying epsilon from 0.0001 to 0.2× edge length changes Room metrics by <0.01% and other metrics by less than 0.05%, confirming stability and negligible impact on accuracy. This threshold-based approach offers a practical trade-off between theoretical rigor and engineering feasibility.

**Computational cost analysis.** We analyze the computational cost of our model with respect to the number of tokens used in the decoder. The metrics include the number of parameters, floating-point operations (FLOPs), inference time per image, and GPU memory usage for a batch of 10. Figure 13 summarizes these metrics for different token counts. Increasing the number of tokens leads to a higher number of parameters and FLOPs, longer inference time, and increased GPU memory consumption. Our method is inherently scalable and built on a general framework. The primary constraint is token length, which is bounded by GPU memory. Scaling is feasible by increasing the token count, though it come with quadratic complexity ($O(n^2)$ in both runtime and memory), as is typical for transformer-based models.

**Visual Analysis.** Figure 14 (main paper) and Figure 15 (appendix) illustrate qualitative differences across ablation settings. Models with edge-based representation and denoising exhibit better closure, alignment, and regularity, even under occlusion or missing data.

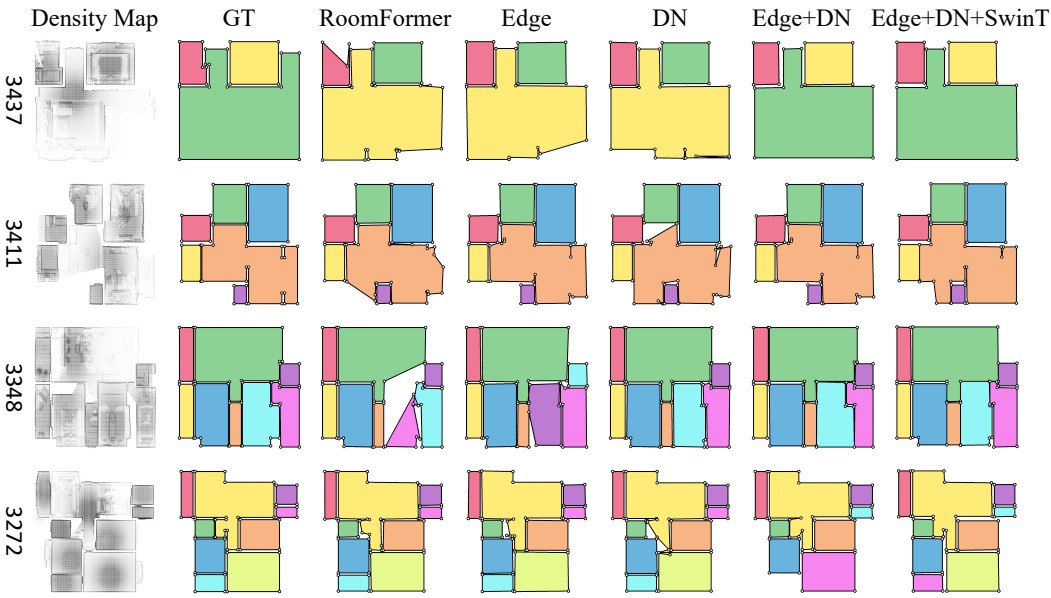

Figure 14: The ablation study on key components. RoomFormer serves as the baseline model. We progressively introduce the Edge module, the Denoising (DN) strategy, and the SwinTransformer-V2 (SwinT) backbone to assess their individual and combined effects.

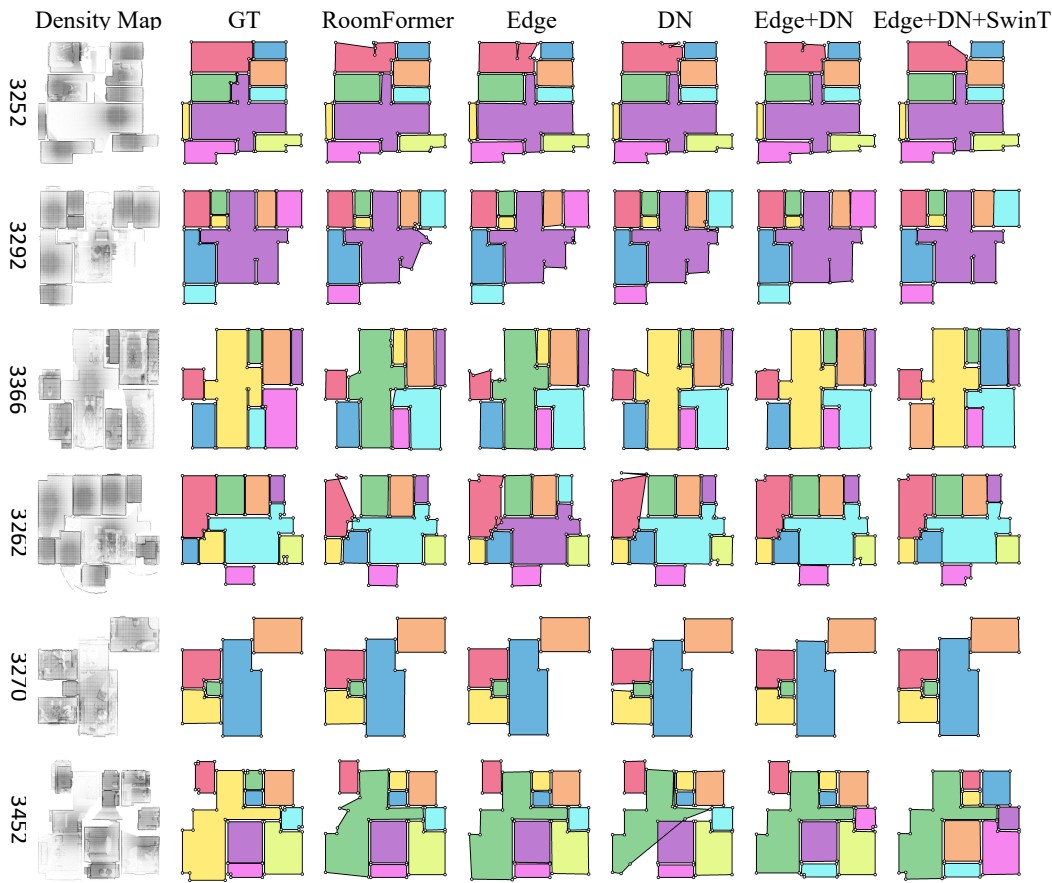

Figure 15: Additional results for The ablation study on key components. RoomFormer serves as the baseline model. We progressively introduce the Edge module, the Denoising (DN) strategy, and the SwinTransformer-V2 (SwinT) backbone to assess their individual and combined effects.

