# OpenReview forum: "CAGE: Continuity-Aware edGE Network Unlocks Robust Floorplan Reconstruction"
_NeurIPS.cc/2025/Conference — NeurIPS 2025 poster_

### Official Review · Reviewer_yHWf · 2025-06-08

**Clarity:** 4
**Significance:** 2
**Originality:** 2
**Rating:** 4
**Confidence:** 3

**Summary:**

The paper addresses the problem of floor plan prediction from a point cloud, i.e., estimate 2D polygons that fit rooms' layout.
The proposed method belongs to the category of methods where the point cloud is pre-processed into a 2D ground density map by projecting the point cloud onto the "ground", which is fed to a network that outputs the floor plan room / polygons.

The contribution of the paper lies in the choice network architecture and training representations:
- polygon are represented by their edges, which detection fails more gracefully than the vertex-based polygon representations used in previous work
- the network is a DN-DETR transformer network

The experiments follow the standard guidelines to evaluate floorplan reconstruction (datasets, metrics) and the baselines are relevant.
The results show that the proposed method achieves better floorplan reconstruction and supports the relevance of the proposed method.

The paper is closely related to RoomFormer  with the following differences:
- RoomFormer has a vertex-based representation whereas CAGE has an edge-based one
- CAGE adopts DN-DETR instead of RoomFormer's DETR

**Questions:**

There is no questions as the paper is very well presented.

**Ethical Concerns:**

["NO or VERY MINOR ethics concerns only"]

**Final Justification:**

Thank you for the rebuttal and for clarifying the importance of the contributions.
Although the paper demonstrates that edges are a sound representation for floor-plan reconstruction, the paper remains very close to RoomFormer, which motivates the rating update from reject to only borderline accept.

**Limitations:**

No, the authors have not adequately addressed the limitations of their work.
Some limitations worth mentioning are:
-  the maximum number of room and the maximum number of edges per room are harcoded in the network so any deployment to spaces with more rooms or room with more edges would require retraining
- the size of the floorplan density map is fixed to 256 x 256 pixels so the network can probably not scale to large spaces as the density map's resolution would decrease, but this is a limitation common to all methods and can be mitigated, e.g.,  by splitting the space into more smaller areas.

There is no potential negative societal impact in this work.

### Misc comments.

A few typos:
- L118: Fig4 -> Fig3
- L160: "N the number of edges" -> "N the maximum number of edges"
- L199: "dummy" -> "mock"
- L243: "Adam" optimizer is missing citation
- L250: "fasted" -> "faster"
- A few undefined references to supp. material marked with "??"

**Quality:**

3

**Strengths And Weaknesses:**

- S1 - The choice of architecture, geometric representation and losses is sound.

- S2 - The related work is exhaustive although it does not highlight the difference with the proposed method enough.

- S3 - The paper is extremely well written, clear and pleasant to read.

- S4 - The experiments follow the standard (datasets, metrics, baseline) and support the relevance of the method.

- W1 - The contribution with respect to RoomFormer is too low for a Neurips submission: the differences are the change of decoder architecture (CAGE adopts the existing DN-DETR instead of RoomFormer's DETR) and the representation of the polygons by their edges. This is what motivates the low rating of the paper.

---

> ### Author Rebuttal · Authors · 2025-07-28
>
> Thank you for your constructive feedback. We will carefully revise the paper to address your concerns and sincerely appreciate your thoughtful review and critical insights.
>
> **Question 1**:  Contribution with respect to RoomFormer
>
> **Answer**:
> - CAGE introduces two key contributions: (1) an edge-based polygon representation that explicitly encodes geometry and direction, enhancing robustness to occlusion and missing corners (e.g., +7.7% Angle F1); and (2) a dual-query denoising transformer decoder, which extends DN-DETR to sequential polygon prediction via perturbed edge queries. Ablation results (Appendix B.5) show that removing either component leads to significant drops (–5.8% Corner F1, –1.9% Angle F1).
> - This is far from a trivial extension. Our edge-centric formulation directly addresses core limitations of corner-based approaches and ensures topological consistency. As recognized by R#fJqK, "the edge representation encodes directionality explicitly and promotes angular regularity… beneficial in floorplan reconstruction." R#Sxuu called it a "simple but efficient change… maintaining directional and structural consistency." Besides, R#EK67 affirmed that "The dual-query decoder with denoising supervision is both novel and effective… robust under occlusion." Notably, while DN-DETR is an existing method, we are the first to adapt it for floorplan reconstruction task. By leveraging denoising supervision, we obviously enhance both accuracy and stability, making this a highly valuable contribution.
> - Together, these innovations yield significant accuracy and robustness gains. We establish a foundation for broader raster-to-vector tasks, such as façade modeling and cadastral map reconstruction, where we are actively extending our framework.
>
> **Question 2**:  Hardcoded maximum number of rooms and edges
>
> **Answer**:
> - Our method is inherently scalable and built on a general framework, with the primary constraint being token length, which is limited by available GPU memory. We use a configuration of 800 tokens (40 polygons × 20 edges), which is sufficient for most residential layouts. For example, Structured3D typically contains fewer than 10 rooms per suit, and SceneCAD includes only one room per case. As demonstrated in case 3410 (Fig. 10, supplementary), our model handles complex multi-room layouts effectively.
> - For larger scenes, our approach can be naturally extended by tiling the input into smaller subregions and merging the outputs—an effective strategy in applications such as cadastral mapping and topographic reconstruction. This allows flexible scaling without requiring retraining or modification of the core architecture.
>
> **Question 3**:  Fixed floorplan density map size
>
> **Answer**:
> - We appreciate the suggestion regarding image tiling. The fixed 256×256 resolution is a common design choice in generative models and has proven sufficient for typical residential layouts in datasets like Structured3D and SceneCAD. For larger or more complex spaces, the resolution can be increased or the floorplan can be tiled into smaller subregions—both are practical, widely used strategies that preserve the core architecture.
>
> **Question 4**:  Typos and Mis-links
>
> **Answer**:
> - Thank you for pointing these out. We will correct all typos and fix the broken references in the revised manuscript.

---

> ### Comment · Reviewer_yHWf · 2025-08-05
> **Rebuttal is relevant and answers technical questions**
>
> Thank you for the rebuttal and for clarifying that some of the references in the review are already in the paper.
>
> The only limitation of the paper is related to the "amount of contributions" with respect to the closest work "RoomFormer" that are the choice in i) edge representation; ii) network architecture.
>
> The rebuttal highlights how this is contribution from related work but this was already very well written in the paper.
>
> This relatively small limitation is compensated for by the overall quality of it (soundness of the methods, experiments, clarity), so the final rating from reject to borderline accept to represent this final assessment.

---

> > ### Author Response · Authors · 2025-08-09
> >
> > Thanks for your valuable comments and suggestions. The manuscript will be revised to address all points raised in our rebuttal. Thanks again for your efforts and feedback!

---

### Official Review · Reviewer_fJqK · 2025-06-29

**Clarity:** 4
**Significance:** 3
**Originality:** 2
**Rating:** 5
**Confidence:** 5

**Summary:**

This paper proposes a method, called CAGE to reconstruct vector floorplans from point cloud density maps. CAGE builds on top of RoomFormer and mainly introduces two designs: (1) Modify the corner representation of each query in RoomFormer to edge representation (i.e. two end points). (2) Add a denoising (DN) training strategy. In addition, the paper explored the importance of backbones (ResNet vs SwinTransformer) in floorplan reconstruction. After integrating all the designs, it attains state-of-the-art results on Structured3D and near-sota results on SceneCAD.

**Questions:**

- I look forward to the authors’ response on weakness 1.

More questions:

- In the edge representation, each query represents two end-points of one edge.
    - How do you decide the order of the two end-points in this edge, i.e. which one is the start/end?
    - Do you think the edge representation actually encodes redundant information? Because two adjacent edges share the common point, that means one point is encoded twice in two edges.
    - As shown in Figure 3, the network cannot predict perfectly shared vertice of two adjacent edges (thus manual merging is needed). Curious question/suggestion: will adding some explicit regularization loss, e.g. forcing the predicted shared vertice of two adjacent edges to be the same, help?
- In tab 4 in the supplementary material, the paper shows incremental ablation on top of RoomFormer. However, it was mentioned that “RoomFormer [12] results are reproduced using our training setup for fair comparison”. But RoomFormer results here are actually worse than in tab. 1. This looks strange to me. Since the proposed method is built on top of RoomFormer, why didn’t you use their training setup to have a strong staring point?

**Ethical Concerns:**

["NO or VERY MINOR ethics concerns only"]

**Final Justification:**

Thank the authors for the detailed and clear rebuttal, which addressed almost all of my concerns. Therefore, I would like to increase my rating from "Borderline accept" to "Accept". I hope the authors will revise the paper accordingly as what they promised in the rebuttal.

**Limitations:**

yes

**Quality:**

3

**Strengths And Weaknesses:**

**Strengths:**

- **The most interesting and novel part is that the paper proposes edge-centric presentation to model floorplan. The motivation makes sense to me.** Compared with RoomFormer’s corner representation, where direction (or connectivity) of corners is implicitly learned using attention, the edge representation encodes directionality explicitly and promotes angular regularity (also reflected on the big improvements on angle metrics). However, I think this representation also has its own limitation, see weaknesses part.
- In addition to regular reconstruction loss adopted from RoomFormer, this paper proposes to add a denoising training objective, where some controlled noises are added to the ground truth and let the network to learn to denoise it. Although this idea is adopted from DN-DETR, **it is interesting to see this strategy is proven beneficial in the floorplan reconstruction task**.
- The paper conducts ablation studies on the two core designs (edge representation and denoising loss) and backbones, **making it clear where the improvements come from**.
- The paper conducts comprehensive comparison with previous methods and achieves state-of-the-art results on Structured3D and near-sota results on SceneCAD.
- The paper contains nice and clear figures, which helps illustrate the designs. The writing is clear and easy to follow.

**Weaknesses:**

- The paper proposes edge-centric representation. Although it is proven superior, **I think the limitation is that it requires heuristics-driven post-processing to merge the predicted edges to final polygons (Figure 3)**. By contrast, RoomFormer directly predicts ordered corner sequences, forming polygons without any post-processing. **In that sense, the proposed edge representation is not elegant to me and I don’t think the paper can claim itself end-to-end floorplan reconstruction anymore, since it actually requires post-processing**.
- This paper conducts ablation studies but puts all of the tables and detailed discussion in the supplementary. However, **I think those experimental results are very important and they should be placed in the main paper**.
- In addition to the two main designs, another difference between the proposed method and RoomFormer is the backbone network. For tab.1 and tab. 2, **I suggest to add another column to indicate the backbone (and number of parameters) of each method. This will make the comparison more fair**.
- Typos:
    - L118: Figure 4(c) → Figure 2(c)
    - L244, L301, L302, L310: “??” missed reference.
    - L528: (main paper) → (appendix)
    - L260: fasted

---

> ### Author Rebuttal · Authors · 2025-07-28
>
> Thank you for your valuable comments and suggestions. We will revise the paper accordingly and truly appreciate your recognition and encouragement.
>
> **Question 1**:  Edge-centric representation and post-processing
>
> **Answer**:
> - Thank you for raising this point. While CAGE performs a lightweight edge-to-polygon conversion, it is not an optimization based post-processing step in the traditional sense. The transformer decoder directly predicts directed edge sequences that form closed loops, and the conversion simply connects these edges into polygons deterministically. It does not involve learning, optimization, or iterative refinement.
> - Importantly, this conversion is highly robust: in the Structured3D dataset, over 98.71% of intersections are of Types I–III (i.e., directionally valid and unambiguous). The conversion's sensitivity to the threshold (epsilon) is minimal—as shown in the table below, varying epsilon from 0.0001 to 0.2× edge length changes Room metrics by <0.01% and other metrics by less than 0.05%, confirming stability and negligible impact on accuracy. This threshold-based approach offers a practical trade-off between theoretical rigor and engineering feasibility.
>
> | eps     | Room Prec. | Room Rec. | Room F1 | Corner Prec. | Corner Rec. | Corner F1 | Angle Prec. | Angle Rec. | Angle F1 |
> |---------|------------|-----------|---------|--------------|-------------|-----------|-------------|------------|----------|
> | 0.2     | 99.59      | 98.69     | 99.14   | 95.04        | 88.57       | 91.70     | 92.57       | 86.40      | 89.39    |
> | 0.1     | 99.59      | 98.69     | 99.14   | 94.99        | 88.55       | 91.66     | 92.51       | 86.37      | 89.34    |
> | 0.05    | 99.59      | 98.69     | 99.14   | 94.96        | 88.55       | 91.65     | 92.46       | 86.34      | 89.30    |
> | 0.01    | 99.59      | 98.69     | 99.14   | 94.95        | 88.58       | 91.66     | 92.40       | 86.34      | 89.27    |
> | 0.0001  | 99.59      | 98.69     | 99.14   | 94.94        | 88.60       | 91.67     | 92.38       | 86.34      | 89.26    |
>
> **Question 2**:  Ablation studies in supplementary
>
> **Answer**:
> - We appreciate your suggestion and will move the key ablation results from Appendix B.5 to the main paper to clearly highlight the impact of edge representation and denoising decoder.
>
> **Question 3**:  Backbone network comparison
>
> **Answer**:
> - Thanks for your suggestion. We will clarify this in the paper and include each model’s backbone and parameter count in Tables 1 and 2. All our baseline results (for RoomFormer and others) use a ResNet-50 backbone. Our method also uses ResNet-50 in these comparisons. CAGE has 40.96M parameters with a ResNet-50 backbone (comparable to RoomFormer’s 40.60M) and 210M with Swin Transformer V2. Despite this, its lightweight architecture enables fast inference (~0.01s/image), matching RoomFormer and outperforming other baselines in speed.
>
> **Question 4**:  Edge endpoint order
>
> **Answer**:
> - We enforce a consistent edge ordering by starting from the top-left corner of each room polygon, listing edges counter-clockwise, with the first endpoint as the starting point. This sequence is used throughout training to ensure consistency.
>
> **Question 5**:  Redundancy in edge representation
>
> **Answer**:
> - Thank you for the insightful suggestion. We experimented with an explicit co-point regularization loss that encourages adjacent edges to share endpoints. The results (see table below) show modest gains in most metrics, particularly for corner and angle F1 scores, though room recall slightly drops. Without this term, our vertex coordinate loss already implicitly promotes shared vertices by aligning predicted edges to ground truth. We will explore this mechanism behind the interesting result in future work.
> - Notably, the explicit regularization improves learning but cannot guarantee exact vertex alignment due to its soft constraint—so edge intersection is still needed during edge-to-polygon conversion.
> - We do not consider the edge representation redundant. Each edge encodes position, direction, and length—information not explicitly available in corner-based approaches. This rich geometric encoding enhances global layout reasoning and precision under occlusion.
>
> | Co-Point | Room Prec. | Room Rec. | Room F1 | Room Prec. | Room Rec. | Room F1 | Room Prec. | Room Rec. | F1 |
> |----------|------------|-----------|---------|--------------|-------------|-----------|-------------|------------|----------|
> | ✗        | 99.59      | 98.69     | 99.23   | 93.88        | 88.79       | 91.26     | 89.65       | 84.88      | 87.20    |
> | ✓        | 99.92      | 98.56     | 99.24   | 93.95        | 89.32       | 91.58     | 89.78       | 85.42      | 87.55    |
>
> **Question 6**: RoomFormer results discrepancy
>
> **Answer**:
> - We reproduced RoomFormer using its official code, hyperparameters, and data splits on our hardware. Slight differences are expected due to implementation environments, but the setup ensures fair ablation comparisons. As shown below, our full model significantly outperforms both the original RoomFormer and our replicated baseline (*), validating the effectiveness of our contributions.
>
> | Method      | Room Prec. | Room Rec. | Room F1 | Corner Prec. | Corner Rec. | Corner F1 | Angle Prec. | Angle Rec. | Angle F1 |
> |-------------|------------|-----------|---------|--------------|-------------|-----------|-------------|------------|----------|
> | RoomFormer  | 97.9       | 96.7      | 97.3    | 89.2         | 85.3        | 87.2      | 83.0        | 79.6       | 81.3     |
> | RoomFormer* | 96.9       | 95.9      | 96.4    | 87.3         | 83.8        | 85.5      | 81.2        | 77.9       | 79.5     |
> | Ours(full)  | 99.6       | 98.7      | 99.1    | 95.0         | 88.6        | 91.7      | 92.5        | 86.4       | 89.3     |
>
> **Question 7**:  Typos and Mis-links
>
> **Answer**:
> - Thank you for pointing these out. We will correct all typos and fix the broken references in the revised version.

---

> > ### Comment · Reviewer_fJqK · 2025-08-03
> > **Thanks for the rebuttal**
> >
> > Thank the authors for the detailed and clear rebuttal, which addressed almost all of my concerns. Therefore, I would like to increase my rating from "Borderline accept" to "Accept". I hope the authors will revise the paper accordingly as what they promised in the rebuttal.
> >
> > One more thing I missed to bring up in my original review but this will not affect my rating - it can be seen as a suggestion for future work: RoomFormer paper shows that their method can be easily extended to reconstruct semantic room types and architectural elements like doors and windows. I think this extension is also applicable to your work. In particular, windows and doors are naturally edges. It would be interesting to see whether the proposed edge representation would benefit the semantic floorplan reconstruction. If so, it would be more valuable to the field.

---

> > > ### Author Response · Authors · 2025-08-05
> > >
> > > Thanks for the recognition of our work and the insightful suggestions. The manuscript will be revised to address all points raised in the rebuttal. As recommended, our future work will extend the scope to include door and window reconstruction. We plan to represent each door / window as a single line segment for geometric reconstruction. Thanks again for your efforts and feedback!

---

### Official Review · Reviewer_EK67 · 2025-07-02

**Clarity:** 2
**Significance:** 3
**Originality:** 3
**Rating:** 5
**Confidence:** 1

**Summary:**

This paper presents CAGE, an end-to-end framework for vectorized floorplan reconstruction from 3D point clouds. The key innovation lies in its edge-centric representation, where each wall is modeled as a directed and geometrically continuous edge, as opposed to traditional corner-based approaches. This design aims to improve robustness under occlusion and noisy input by explicitly encoding structural continuity. The authors introduce a dual-query transformer decoder that integrates both latent and perturbed edge queries within a denoising framework, enabling stable training and iterative polygon refinement. Experiments on Structured3D and SceneCAD demonstrate that CAGE achieves state-of-the-art performance across room, corner, and angle-level metrics, and shows strong cross-dataset generalization.

**Questions:**

See weaknesses

**Ethical Concerns:**

["NO or VERY MINOR ethics concerns only"]

**Limitations:**

See weaknesses

**Quality:**

3

**Strengths And Weaknesses:**

**Strengths**:
- The paper is well-structured and clearly written. It introduces a strong motivation for moving from corner-based to edge-based representations in floorplan reconstruction, and presents the approach in a logically coherent manner with helpful visual illustrations.
- The proposed edge-centric formulation enables topologically consistent and robust polygon reconstruction, especially under partial occlusion or missing data—an important challenge in real-world scans.
- The use of a dual-query transformer decoder with denoising supervision is both novel and effective. It improves convergence and prediction stability, and is well-supported by ablation studies and competitive results on Structured3D and SceneCAD.
- The method achieves state-of-the-art performance on several key metrics (Room, Corner, Angle) and demonstrates strong cross-dataset generalization without post-processing.

**Weaknesses**:
- Although the paper emphasizes robustness under occlusion and clutter, it lacks controlled evaluation under extreme occlusion scenarios (e.g., dense furniture occluding multiple walls). A more detailed analysis of failure cases would strengthen the empirical claims.
- The paper does not evaluate performance on datasets with complex geometries such as curved walls or highly non-Manhattan layouts, despite claiming general applicability. While the authors mention such diversity in Structured3D, there is no specific analysis or visualization to confirm robustness in these cases.
- I am not an expert in floorplan reconstruction, but it seems that the edge-based representation—while robust—might struggle to capture very short wall segments or fine structural details compared to corner-based methods. This tradeoff is not discussed.
- Similarly, the paper does not report model size, parameter count, or compute time beyond per-frame inference speed. From a system perspective, it would be helpful to better understand the computational efficiency. Again, this is an intuition rather than a domain-specific concern.

---

> ### Author Rebuttal · Authors · 2025-07-28
>
> Thank you for your valuable comments and suggestions. We will revise the paper accordingly and truly appreciate your recognition and encouragement.
>
> **Question 1**:  Robustness under extreme occlusion
>
> **Answer**:
> - We evaluated CAGE under severe occlusion scenarios, highlighted by red circles in Fig. 6 and Fig. 11 (supplementary). E.g., in Scene 3292, the living room has two fully unscanned walls and one partially visible wall due to the absence of a scanner. In Scene 3356, two walls in the bottom-right room are unscanned, and one wall in the top-right bedroom is fully blocked by a wardrobe. Despite these challenges, CAGE reconstructs complete room polygons, showing strong robustness where corner-based methods often fail.
>
> **Question 2**: Complex geometries
>
> **Answer**:
> - CAGE is not limited to Manhattan layouts or flat surfaces. As highlighted in red circles in Scene 3493 (Fig. 6) and Scene 3443 (Fig. 10), as well as in many cases in Figs. 7–8, CAGE handles slanted walls by predicting edges at arbitrary angles and approximates curved structures using short segments, similar to other polygon-based methods. While Structured3D and SceneCAD do not contain curved geometry, cross-dataset testing on our collected data confirms CAGE’s generalizability. We will include examples of curved walls in the supplementary material of the revised version.
>
> **Question 3**:  Short wall segments
>
> **Answer**:
> - CAGE captures short wall segments and fine structures effectively, including alcoves and dividers (e.g., the living room in Fig. 6). Its edge-based representation, combined with global reasoning, reduces the risk of missing details that corner-based methods often overlook. Even at 256×256 resolution, CAGE remains robust to noise and occlusion, with potential for further gains at higher resolutions.
>
> **Question 4**:  Computational efficiency
>
> **Answer**:
> - Thank you for the suggestion. We will report full efficiency details in the revised manuscript. CAGE has 40.96M parameters with a ResNet-50 backbone (vs. RoomFormer’s 40.60M), converges stably (650 epochs on Structured3D, 400 on SceneCAD), and achieves fast inference (~0.01s/image), matching RoomFormer and outperforming other baselines in speed.

---

> > ### Author Response · Authors · 2025-08-05
> >
> > Thank you very much for taking the time to review our submission and provide your evaluation. If there are any remaining concerns or questions that were not addressed in our previous response, please let us know, we would be happy to further clarify or provide more information. Thank you again for your efforts and feedback!

---

### Official Review · Reviewer_SXuu · 2025-07-02

**Clarity:** 3
**Significance:** 3
**Originality:** 3
**Rating:** 5
**Confidence:** 4

**Summary:**

This paper deals with floor-plan reconstruction from 3D point clouds; such a direction is a longstanding goal in the computer vision domain especially for navigation and AR simulation tasks. Instead of relying on a corner-based or line-grouping technique which can be sensitive to noisy data, the authors introduce an edge-based polygon formulation to model each wall as a directed and geometrically continuous edge. CAGE builds a transformer decoder-based denoising framework upon this representation to generate topologically valid room layouts. With experiments on the current publicly available datasets and comparisons with recent state-of-the-art methods, this work makes a strong case for resilience to occlusion and layout noise in floor-plan reconstruction.

**Questions:**

- As mentioned in the Weaknesses section, I would like to see the answers.
- Why is there extreme variability in different checkpoints within the same training run? Is this due to convergence related issues of the model?
- How does multi-room floor-plan estimation work? Some visual examples would be helpful
- Suggestion: the authors can incorporate training on synthetic datasets such as Aria Synthetic Environments (ASE) to improve generalization precision

I look forward to the authors' responses on the rebuttal and discussion phase.

**Ethical Concerns:**

["NO or VERY MINOR ethics concerns only"]

**Final Justification:**

All my and most other reviewers' concerns were clearly addressed in detail in the rebuttal. I keep my score at Accept. I hope the authors will revise the manuscript with the changes from the rebuttal.

**Limitations:**

Yes

**Paper Formatting Concerns:**

None.

**Quality:**

3

**Strengths And Weaknesses:**

Strengths:
- Technically well-motivated and simple but efficient change from corner to edge-based representation – maintains directional and structural consistency
- Polygon matching with edge regression and rasterization losses are intuitive for training a denoising framework in an end-to-end manner (as also shown in the ablation study)
- Extensive technical validation + comparison with sota methods + detailed limitation analysis

Weaknesses:
- [Table 2 and 3, main paper] Why is PolyRoom so much better on SceneCAD and cross-dataset generalization precision?
- [Figure 3, main paper] How sensitive is the method to distance threshold selection for Type IV and others?
- Any intuition as to why SwinTransformer degrades in performance when used in isolation?
- Extremely limited runtime analysis – does CAGE have the same speed as RoomFormer? What does lightweight architecture and no need for additional refinement mean? [L261-262]
- Since the edge-based representation models room layout as fixed-length sequence, how does this scale with complexity of room layout? And, does this affect performance, both in terms of speed and accuracy?
- [Minor] missing links and references in the draft to figures/tables.

---

> ### Author Rebuttal · Authors · 2025-07-28
>
> Thank you for your valuable comments and suggestions. We will revise the paper accordingly and truly appreciate your recognition and encouragement.
>
> **Question 1**: PolyRoom better on SceneCAD and cross-dataset generalization precision?
>
> **Answer**:
> - PolyRoom builds on RoomFormer and leverages a segmentation module (Mask2Former) to generate initial query contours, making it a non-end-to-end method. Its strong performance on SceneCAD—a single-room dataset—largely benefits from the segmentation model, which already provides high-quality priors by the segmentation module. However, this reliance limits its ability to encode multiple-room layouts or perform global reasoning, often leading to room-level conflicts. While PolyRoom shows higher precision due to its dependence on segmentation quality, our method achieves superior recall and F1 on key metrics (Room IoU, Corner Recall, Angle Recall, and Angle F1), highlighting stronger generalization and structural robustness.
>
> **Question 2**: How sensitive is the method to distance threshold selection for Type IV and others?
>
> **Answer**:
> - The edge-to-polygon convertion is empirically stable with respect to the threshold. Type I intersections account for 97.06%, while Types II–IV cover 1.17%, 0.47%, and 1.29%. Shown in the following table, varying the threshold (0.0001 to 0.2× edge length) changes metrics by less than 0.05%, confirming robustness to the epsilon value.
>
> | eps     | Room Prec. | Room Rec. | Room F1 | Corner Prec. | Corner Rec. | Corner F1 | Angle Prec. | Angle Rec. | Angle F1 |
> |---------|------------|-----------|---------|--------------|-------------|-----------|-------------|------------|----------|
> | 0.2     | 99.59      | 98.69     | 99.14   | 95.04        | 88.57       | 91.70     | 92.57       | 86.40      | 89.39    |
> | 0.1     | 99.59      | 98.69     | 99.14   | 94.99        | 88.55       | 91.66     | 92.51       | 86.37      | 89.34    |
> | 0.05    | 99.59      | 98.69     | 99.14   | 94.96        | 88.55       | 91.65     | 92.46       | 86.34      | 89.30    |
> | 0.01    | 99.59      | 98.69     | 99.14   | 94.95        | 88.58       | 91.66     | 92.40       | 86.34      | 89.27    |
> | 0.0001  | 99.59      | 98.69     | 99.14   | 94.94        | 88.60       | 91.67     | 92.38       | 86.34      | 89.26    |
>
> **Question 3**: Any intuition as to why SwinTransformer degrades in performance when used in isolation?
>
> **Answer**:
> - In our experiments, SwinTransformer captures fine details but is more sensitive to noise, often resulting in irregular or self-intersecting polygons when used directly in the RoomFormer framework. In contrast, our method better leverages Swin’s global reasoning through the edge-based representation and dual-query strategy, enhancing robustness and yielding more stable and accurate layouts.
>
> **Question 4**: Extremely limited runtime analysis – does CAGE have the same speed as RoomFormer? What does "lightweight architecture and no need for additional refinement" mean? [L261-262]
>
> **Answer**:
> - CAGE has 40.96M parameters with a ResNet-50 backbone (vs. RoomFormer’s 40.60M), converges stably (650 epochs on Structured3D, 400 on SceneCAD), and achieves fast inference (~0.01s/image), matching RoomFormer and outperforming other baselines in speed. We will include a detailed runtime analysis in the revised version.
> - The phrase “lightweight architecture and no need for additional refinement” refers to our advantage over all other methods except RoomFormer. CAGE only performs a simple edge-to-polygon conversion, making it faster than models requiring complex refinement or optimization.
>
> **Question 5**: Since the edge-based representation models room layout as fixed-length sequence, how does this scale with complexity of room layout? And, does this affect performance, both in terms of speed and accuracy?
>
> **Answer**:
> - We use a configuration of 800 tokens (40 polygons × 20 edges), which is sufficient for most residential layouts. For example, Structured3D typically contains fewer than 10 rooms per floor, and SceneCAD includes only one room per case. As demonstrated in case 3410 (Fig. 10, supplementary), our model handles complex multi-room layouts effectively.
> - Our method is inherently scalable and built on a general framework. The primary constraint is token length, which is bounded by GPU memory. Scaling is feasible by increasing the token count, though it come with quadratic complexity (O(n²) in both runtime and memory), as is typical for transformer-based models.
> - For very larger scenes, our approach can be naturally extended by tiling the input into smaller subregions and merging the outputs—a widely used and effective strategy in applications such as cadastral mapping and topographic reconstruction.
>
> | Number of Tokens | Params(M) | FLOPs(G) | inference time (ms/image) | 10*GPU（1Batch） |
> |------------------|-----------|----------|---------------------------|-----------------|
> | 1200             | 41.1      | 26.4     | 4.7                       | 12.6            |
> | 1400             | 41.1      | 29.7     | 5.5                       | 15.9            |
> | 1600             | 41.2      | 33.2     | 6.1                       | 19.8            |
> | 1800             | 41.2      | 37.0     | 7.2                       | 24.1            |
> | 2000             | 41.3      | 41.1     | 8.1                       | 28.8            |
> | 2200             | 41.3      | 45.4     | 9.1                       | 34.1            |
> | 2400             | 41.4      | 50.0     | 10.2                      | 39.8            |
> | 2600             | 41.4      | 54.9     | 11.5                      | 46.0            |
> | 2800             | 41.5      | 60.0     | 12.8                      | 52.6            |
> | 3000             | 41.5      | 65.4     | 14.2                      | 59.8            |
> | 3200             | 41.6      | 71.0     | 15.6                      | 67.4            |
>
> **Question 6**: Why is there extreme variability in different checkpoints within the same training run? Is this due to convergence related issues of the model?
>
> **Answer**:
> - We observed some variability across checkpoints, likely due to stochastic elements such as dropout. However, these fluctuations are minor, and overall convergence is stable. We repeatedly verified the experiments, and the results reported in the paper are consistent and robust across runs.
>
> **Question 7**: How does multi-room floor-plan estimation work? Some visual examples would be helpful.
>
> **Answer**:
> - We adopt a two-level token structure where tokens 0–m represent the first polygon, m+1–2m the second, and so on, allowing structured and efficient multi-room representation. We will include additional visual examples to better illustrate this in the revised version.
>
> **Question 8**: Suggestion: the authors can incorporate training on synthetic datasets such as Aria Synthetic Environments (ASE) to improve generalization precision.
>
> **Answer**:
> - Thank you for the suggestion. ASE’s large-scale synthetic dataset (100K scenes, ~28× larger than Structured3D) is indeed valuable. We are actively collecting diverse real and synthetic scenes, including ASE, to further enhance generalization in future extensions.
>
> **Question 9**: Typos and Mis-links:
>
> **Answer**:
> - Thank you for pointing these out. We will correct all typos and fix the broken references in the revised version.

---

> > ### Comment · Reviewer_SXuu · 2025-08-04
> >
> > Thanks to the authors for their efforts on the rebuttal. All my and most other reviewers' concerns were clearly addressed in detail. I keep my score at Accept. I hope the authors will revise the manuscript with the changes from the rebuttal.
> >
> > PS: I also agree with Reviewer fJqK that extending this method to semantic room types would be an extremely interesting application, in the context of current AR/VR scenarios.

---

> > > ### Author Response · Authors · 2025-08-05
> > >
> > > Thanks for the recognition of our work and the insightful suggestions. The manuscript will be revised to address all points raised in the rebuttal. As recommended, our future work will extend the scope to include door and window reconstruction. We plan to represent each door / window as a single line segment for geometric reconstruction. Thanks again for your efforts and feedback!

---

### Decision · Program_Chairs · 2025-09-17

**Decision:**

Accept (poster)

**Comment:**

The paper describes a new system called CAGE (Continuity-Aware edge) for floor-plan reconstruction from a 3D point cloud (via projection onto a 2D density map). Walls are modeled as geometrically continuous edges, with each room represented by a (watertight) polygon. The architecture is based on a dual-query transformer decoder.

Pros: The paper is well-written, with the novelty being the robust edge-centric approach to modeling floorplan. While there are ideas adapted from RoomFormer and DN-DETR, the unique design decisions are highly effective in generating SOTA (or near SOTA) results for Structured3D and SceneCAD.

Cons: The system is not exactly end-to-end because post-processing is required. In addition, the maximum number of rooms and edges per room are hardcoded, which limits practicality of the system.

The novelty in representing the floorplan and architectural design decisions as well as the SOTA results are the main reasons for acceptance. There is consensus among the reviewers for accepting this paper.